# Restrictive Atrial Dysfunction in Cardiac Amyloidosis: Differences between Immunoglobulin Light Chain and Transthyretin Cardiac Amyloidosis Patients

**DOI:** 10.3390/biomedicines10081768

**Published:** 2022-07-22

**Authors:** Mathijs O. Versteylen, Maaike Brons, Arco J. Teske, Marish I. F. J. Oerlemans

**Affiliations:** Department of Cardiology, University Medical Center Utrecht, P.O. Box 85500, 3508 GA Utrecht, The Netherlands; mathijs.versteylen@gmail.com (M.O.V.); m.brons@umcutrecht.nl (M.B.); a.j.teske-2@umcutrecht.nl (A.J.T.)

**Keywords:** cardiac amyloidosis, restrictive atrial dysfunction, atrial strain, ATTR, AL

## Abstract

**Background:** In cardiac amyloidosis, the prevalence of thromboembolic events and atrial fibrillation is higher in transthyretin amyloidosis compared to immunoglobulin light chain amyloidosis. Therefore, we hypothesize that transthyretin cardiac amyloidosis patients have worse atrial function. **Purpose:** To explore the left atrial function by conventional ultrasound and strain analysis in immunoglobulin light chain- and transthyretin cardiac amyloidosis patients. **Methods:** In cardiac amyloidosis patients in our Amyloidosis Expert Center, echocardiographic strain analysis was performed using speckle tracking. **Results:** The data of 53 cardiac amyloidosis patients (83% male, mean age 70 years) were analyzed. Transthyretin cardiac amyloidosis patients (*n* = 24, 45%) were older (75 ± 5.6 vs. 65 ± 7.2 years, *p* < 0.001) and had more left ventricular (LV) hypertrophy than immunoglobulin light chain cardiac amyloidosis patients (*n* = 29, 55%). However, LV systolic and diastolic function did not differ, nor did left atrial dimensions (LAVI 56(24) vs. 50(31) mL/m^2^). Left atrial reservoir strain was markedly lower in transthyretin cardiac amyloidosis (7.4(6.2) vs. 13.6(14.7), *p* = 0.017). This association was independent of other measurements of the left atrial and ventricular function. **Conclusions:** Transthyretin cardiac amyloidosis patients had lower left atrial reservoir function compared to immunoglobulin light chain cardiac amyloidosis patients although the left atrial geometry was similar. Interestingly, this association was independent of left atrial- and LV ejection fraction and global longitudinal strain. Further research is warranted to assess the impact of impaired left atrial dysfunction in transthyretin cardiac amyloidosis on atrial fibrillation burden and prognosis.

## 1. Introduction

Amyloidosis is a systemic, life-threatening, and progressive disease characterized by the deposition of amyloid fibrils from previously soluble precursor proteins that have become insoluble after misfolding. In the heart, the accumulation of amyloid fibrils leads to increased diastolic chamber stiffness and ventricular wall thickening. Ultimately, this will result in a restrictive cardiomyopathy with distinct ventricular diastolic dysfunction [1]. The main types of cardiac amyloidosis are immunoglobulin light chain (AL) and transthyretin (ATTR) amyloidosis. AL amyloidosis is caused by an underlying plasma cell dyscrasia, leading to extensive light chain production and amyloid formation. ATTR cardiac amyloidosis is caused by amyloid originating from transthyretin, which has become unstable due to aging in wild-type (normal) transthyretin amyloidosis (ATTRwt) or due to a genetic transthyretin mutation, leading to a variant transthyretin protein (ATTRv). Survival is dependent on the disease stage at diagnosis, and in patients with advanced heart failure, the median survival for AL cardiac amyloidosis is 6–12 months [2]. For ATTR cardiac amyloidosis, the median survival is 2–6 years, although these numbers are based on data before the introduction of disease-modifying drugs [3].

In the past years, several novel treatments have become available including transthyretin stabilizers and transthyretin gene silencing approaches for ATTR cardiac amyloidosis [1]. Moreover, the treatment for AL cardiac amyloidosis has improved significantly over the past decade [4]. Nevertheless, the treatment for both types of cardiac amyloidosis is most beneficial in patients presenting in an early stage of the disease. Unfortunately, the diagnosis of cardiac amyloidosis is often delayed [5,6]. Early identification is hampered by the unawareness of cardiac amyloidosis by cardiologists and other physicians, misconceptions regarding the diagnosis, the heterogenic multi-systematic nature of cardiac amyloidosis, and the difficulties in obtaining pathological confirmation [7,8]. Furthermore, early stage cardiac amyloidosis is mostly characterized by nonspecific symptoms and current biomarkers lack specificity. Many imaging characteristics of cardiac amyloidosis have been found in an advanced stage of the disease, making early diagnosis difficult [9]. Recently, we described the automated identification of patients with unexplained left ventricular hypertrophy including cardiac amyloidosis, using text mining and machine learning [10]. Interestingly, another group developed a machine learning algorithm where atrial fibrillation was shown to be a pivotal factor in the identification of cardiac amyloidosis patients [11]. It is known that cardiac amyloidosis is characterized by a high prevalence of atrial fibrillation and a high thromboembolic event rate [12]. In up to 70% of the patients with ATTR cardiac amyloidosis, atrial fibrillation is present, which is higher than in AL cardiac amyloidosis patients [13,14]. In both AL and ATTR cardiac amyloidosis patients, the presence of atrial thrombi seems to be high [15]. The exact role of atrial function in the risk of thromboembolism is currently unknown in cardiac amyloidosis. In the general population, left atrial strain was reduced in patients with atrial fibrillation with low CHADS2 score having stroke [16]. Additionally, in a population of chronic atrial fibrillation with a low number of patients on oral anticoagulation, left atrial reservoir strain predicted stroke, independent of atrial volume, E/e’, or CHAD2 score [17]. The apparent potential of atrial function measurement in thromboembolic risk stratification could have clinical implications for the consideration of anticoagulation in cardiac amyloidosis patients.

Although the left atrial dimension is a broadly used marker with prognostic value [18,19], it lacks information about the left atrial function. Therefore, it is predominantly a marker of long standing high filling pressures, instead of an early marker of atrial dysfunction, as seen in amyloid deposition. In cardiac amyloidosis, atrial functional decline beyond diastolic dysfunction seems to be a result of the direct amyloid infiltration of the atrium. Namely, both fibrosis as visualized on cardiac magnetic resonance imaging [20] as well as histological amyloid infiltration of the atrium is associated with poor atrial function [21,22]. In addition, atrial function is associated with atrial fibrillation in different populations. For instance, left atrial function predicted the occurrence of atrial fibrillation in an elderly population [23] as well as recurrent atrial fibrillation after electrical cardioversion [24]. Left atrial strain measurements have also been shown to provide independent prognostic value in predicting cerebrovascular events in atrial fibrillation patients [16,17,25].

As amyloidosis progressed, the left atrium showed increased amyloid deposition and fibrosis [21], resulting in a loss of compliance and contractile function [22]. Left atrial function measured by strain imaging therefore seems to serve as a proxy for left atrial fibrosis caused by amyloid deposition. Together with altered calcium handling, atrial fibrosis is an important causative substrate of atrial fibrillation [26,27]. Recent studies in amyloidosis patients have shown that left atrial dysfunction is associated with all-cause mortality [21,28]. Due to the significant diastolic dysfunction in cardiac amyloidosis, the loss of the atrial contribution to ventricular filling is hemodynamically poorly tolerated and is associated with clinical deterioration and rehospitalization [15]. Interestingly, the absence of atrial mechanical contraction in sinus rhythm had similar prognostic implications compared to atrial fibrillation, probably underlining the importance of atrial function [27].

The left atrium has a reservoir (expansion during ventricular systole), conduit (early diastolic emptying), and booster pump function (atrial contraction), which can be identified using deformation strain imaging [29]. In particular, the reservoir strain function is known to decline in cardiac amyloidosis [30,31]. This atrial restrictive dysfunction might be one of the imaging hallmarks in cardiac amyloidosis and could play an important role in the early detection of the disease. However, whether this is useful for both AL and ATTR amyloidosis remains unclear. To provide more insight, we performed a detailed characterization of the left atrial function by conventional ultrasound and strain analysis in immunoglobulin light chain- and transthyretin cardiac amyloidosis patients.

We hypothesize that ATTR cardiac amyloidosis is accompanied by a more pronounced atrial dysfunctional compared to AL amyloidosis.

## 2. Methods

### 2.1. Study Population

In this single-center retrospective study, patients referred to the Amyloidosis Expert Center Utrecht, the Netherlands, analyzed using General Electric (GE) echo machines, were included from 2017 until 2020. Patients studied with other vendors were excluded to correct for inter-vendor variability (especially as speckle tracking analysis is known to be variable among vendors). All of the included patients provided written informed consent, and the study was approved by the Medical Ethics Committee of the University Medical Center Utrecht (non-WMO 19/222) and conducted in accordance with the Declaration of Helsinki. Demographics and clinical data were collected from the electronic health records of patients with a final diagnosis of ATTR cardiac amyloidosis (wild-type and hereditary) or AL cardiac amyloidosis who were aged 18 years or older. Occurrence of atrial fibrillation was registered within a two year window (e.g., within 2 years prior to 2 years after initial echocardiographic study), using follow-up electrocardiogram studies.

### 2.2. Diagnosis of Cardiac Amyloidosis

Patients referred to the Amyloidosis Expert Center Utrecht with cardiac signs and/or symptoms for the suspicion of cardiac amyloidosis were invited for a first consultation in the expert outpatient clinic with a cardiologist and/or hematologist and/or neurologist specialized in amyloidosis. Next to medical history taking, patients underwent physical examination. All patients underwent cardiac evaluation electrocardiogram (ECG), echocardiogram and biochemistry (e.g., N-terminal prohormone of brain natriuretic peptide (NT-proBNP), Troponin, and estimated glomerular filtration rate (eGFR) to assess cardiac involvement. The final diagnosis of AL cardiac amyloidosis was established when systemic AL amyloidosis was confirmed by positive Congo red staining of abdominal fat pad biopsy or targeted organ biopsy, further demonstrated by a positive immunohistochemistry for kappa or lambda light chain and a negative staining for amyloid type A (AA) and ATTR cardiac amyloidosis. For ATTR cardiac amyloidosis, the final diagnosis was established by a positive bone scintigraphy (grade II or III) in the absence of a plasma cell dyscrasia. Endomyocardial biopsy was only performed in two patients where non-invasive diagnosis of ATTR cardiac amyloidosis could not be established, demonstrated by the positive Congo red and transthyretin staining by immunohistochemistry.

When the cardiologist and/or hematologist and/or neurologist suspected ATTR cardiac amyloidosis, they referred the patient to the clinical geneticist specialized in cardiovascular diseases for genetic screening.

With the use of shared decision making, the cardiologist, hematologist, and neurologist supported the patient to reach a decision about their medical treatment. In multiple disciplinary team meetings, comprehensive treatment plans for patients with cardiac amyloidosis were developed. Most of the patients were referred to their own cardiologist for monitoring and treatment of heart failure with the established treatment plan. Interdisciplinary consults were performed when cardiologists of the referred hospital had any questions concerning the treatment of patients with cardiac amyloidosis. Follow-up visits for cardiac amyloidosis were performed at least one every year at the Amyloidosis Expert Center Utrecht, depending on the severity of cardiac amyloidosis, the treatment plan, and patient preferences.

### 2.3. Echocardiographic Assessment

A complete transthoracic echocardiographic study was performed, according to the recommendations of the European Association of Cardiovascular Imaging [32]. Left atrial volume index (LAVI) was calculated using the two plane area length method for left atrial volume corrected for BSA (Haycock formula). Left ventricular ejection fraction was calculated using semi-automated “Auto-EF” software using biplane contouring. Speckle tracking analysis was used for global longitudinal strain (GLS) of the left ventricle. Left- and right atrial strain was performed on apical four chamber views, and ventricular end-diastole was the time reference for zero atrial strain, following the EACVI consensus document [29]. Echocardiograms were obtained using a commercially available ultrasound machine (Vivid e9 and Vivid e95, GE Healthcare, Horten, Norway). Data were stored and transferred to a computer workstation for offline analysis. All echocardiograms were analyzed by one experienced reader (MV). Conflicting measurements and difficult deformation imaging analysis were re-evaluated by an experienced second reader (AT). Dimensions, velocities, left ventricular ejection fraction, global longitudinal strain, and left atrial and right atrial strain were measured using EchoPAC, version 203 (GE Healthcare, Horten, Norway). All patients had sinus rhythm during transthoracic echocardiography.

### 2.4. Statistical Analyses

Categorical variables were expressed as the number of patients and percentages. Continuous variables were expressed as means with standard deviation (SD), or median with interquartile range (IQR) for non-normal distribution, respectively. The Chi-square test was used for categorical variables, and the independent-samples t-test for continuous variables in the case of normal distributions; the Mann–Whitney U-test in the case of skewed distributions. The multivariate binary logistic regression model was used, and logarithmic transformation was used for parameters with a non-linear distribution. Data were analyzed using SPSS software (IBM SPSS statistics 26), and a significance level of 0.05 was used.

## 3. Results

### 3.1. Patient Population

The study population consisted of 53 patients referred to our Amyloidosis Center from 2017 to 2020 who underwent a complete transthoracic echocardiographic study (Table 1). This study is part of a population previously described [33]. Out of the total population of 55 patients with cardiac amyloidosis, two were excluded because the measurement of the left atrial strain was technically unsuccessful due to insufficient image quality. Mean age was 70 ± 8 years, 44 (83%) were male and 24 (45%) had a final diagnosis of ATTR cardiac amyloidosis, and 29 (55%) patients were diagnosed with AL cardiac amyloidosis. Patients with ATTR cardiac amyloidosis were significantly older, more often had atrial fibrillation, and had lower NT-proBNP levels (ATTR cardiac amyloidosis: 2809 ± 1724 vs. AL cardiac amyloidosis: 5577 ± 5848; *p*-value 0.04). The baseline characteristics are described in Table 1.

### 3.2. Atrial Fibrillation

In 36 patients (68%), atrial fibrillation occurred during the follow-up period. In ATTR cardiac amyloidosis patients, the incidence was higher compared to the AL cardiac amyloidosis patients, 16 (55%) vs. 20 (83%), *p*-value 0.03. A history of atrial fibrillation was already present in 29 (55%) patients (ATTR cardiac amyloidosis, 16 patients vs. AL cardiac amyloidosis, 13 patients). Patients with AL cardiac amyloidosis and a history of atrial fibrillation had more advanced cardiac amyloidosis than the ATTR cardiac amyloidosis patients. None of the patients with ATTR cardiac amyloidosis had advanced cardiac amyloidosis (Gillmore score >II) versus 10 patients (77%) with AL cardiac amyloidosis (MAYO clinic staging system (MAYO) stage III). In this group, the median time of manifestation of atrial fibrillation before cardiac amyloidosis diagnosis did not differ between patients with ATTR cardiac amyloidosis and AL cardiac amyloidosis; 10 (4–22) months vs. 8 (3–24) months, respectively.

In the seven patients (four AL cardiac amyloidosis vs. three ATTR cardiac amyloidosis) with new onset atrial fibrillation after cardiac amyloidosis diagnosis, the median time from the diagnosis of cardiac amyloidosis to the onset of atrial fibrillation was 10 (5–20) months. Although not statistical significant, atrial fibrillation manifested quicker in patients with ATTR cardiac amyloidosis than in patients with AL cardiac amyloidosis (ATTR cardiac amyloidosis patients: 9 (6–13) vs. 17 (17–24) months in immunoglobulin light chain cardiac amyloidosis).

### 3.3. Difference between AL- and ATTR Cardiac Amyloidosis

In general, patients with ATTR cardiac amyloidosis were older, and had lower NT pro-BNP values. Functional capacity did not differ (Table 1). Echocardiographic dimensions differed between patients with AL- and ATTR cardiac amyloidosis (Table 2). Patients with ATTR cardiac amyloidosis had more pronounced left ventricular hypertrophy while the left atrial dimensions did not differ between the groups.

The echocardiographic global systolic and diastolic assessment showed that left ventricular ejection fraction, global longitudinal strain, nor diastolic parameters such as E/e’ and tricuspid regurgitation velocity differed between AL- and ATTR cardiac amyloidosis (Table 3). Interestingly, only the left atrial reservoir and pump strain were significantly different between AL cardiac amyloidosis and ATTR cardiac amyloidosis.

### 3.4. Regression Analysis

We investigated the association with the ATTR cardiac amyloidosis of the left volume and functional parameters, and left ventricular parameters (Table 4). In addition, we investigated the association of the left reservoir function with ATTR cardiac amyloidosis in a multivariate regression model, with other assessments of the left atrial and left ventricular function. The left atrial reservoir strain is associated with ATTR cardiac amyloidosis, independent of left atrial volume index and left atrial ejection fraction, left ventricular ejection fraction, and GLS (B-value 0.16; 95% CI 0.03–0.79; *p*-value 0.02).

## 4. Discussion

In the current single center exploratory study, we sought to evaluate the potential differences in the atrial mechanics, which could explain the clinical observation that atrial fibrillation and systemic thromboembolic complications are more frequent in ATTR cardiac amyloidosis as opposed to AL cardiac amyloidosis. The main findings of our study were as follows: (1) Left atrial reservoir function, as determined by speckle tracking strain analysis, was lower in ATTR cardiac amyloidosis compared to AL cardiac amyloidosis, in a consecutive group of cardiac amyloidosis patients despite comparable left atrial volumes; (2) although ATTR cardiac amyloidosis patients had more pronounced left ventricular hypertrophy compared to AL cardiac amyloidosis, systolic function, as determined by ejection fraction and global longitudinal strain, did not differ; also, the diastolic function parameters were equal; and finally, (3) the association between the left atrial reservoir function and ATTR cardiac amyloidosis was independent of the other measurements of the left atrial dimension and function, and from the left ventricular function. Therefore, the findings in our study suggest that left atrial functional quantification using strain analysis could be a useful parameter in cardiac amyloidosis patients as it is an autonomous marker of atrial restrictive dysfunction, and may provide insights into the disease etiology and its consequences.

In previously published studies in different populations, the relationship between atrial function as assessed by speckle tracking and atrial fibrillation has been reported. For instance, the left atrial reservoir function assessed by strain analysis predicted new-onset atrial fibrillation, independent of the left atrial volume and left ventricular function [23]. It also predicted recurrent atrial fibrillation after electrical cardioversion [24], and latent atrial fibrillation in a large group of embolic stroke with an undetermined source [34]. Therefore, the more pronounced atrial dysfunction might explain the higher incidence of atrial fibrillation in ATTR cardiac amyloidosis [35].

The difference in left atrial function between amyloid light chain- and ATTR cardiac amyloidosis that we report here is in line with previous publications [30], suggesting a different etiology among amyloidosis subtype. Of course, left atrial function is heavily affected by left ventricular function, especially long lasting diastolic dysfunction and elevated filling pressures. Above this, increased left atrial amyloid deposition and fibrosis result in restrictive atrial dysfunction, for example, a loss of compliance and contractile function [21,22]. Our results suggest direct amyloid infiltration of the atrium, leading to the loss of reservoir function and contractility.

In terms of the diagnostic process of cardiac amyloidosis, echocardiography will remain the single most important and widely available imaging technique. Although very advanced cardiac amyloid deposition results in a relatively distinctive cardiomyopathy, the echocardiographic recognition of amyloidosis is often challenging. In particular, early stages of cardiac amyloidosis might be hard to recognize, even for experienced echocardiography readers. The relevance of an early diagnosis, however, is crucial for patient prognosis [36]. Once heart failure symptoms occur, the patient prognosis worsens. This stage can be avoided by initiating early treatment. The hematologic treatment of AL development has shown much progression in the last decade [4]. In addition, therapeutic agents have become available for ATTR cardiac amyloidosis. Both gene silencers for genetic as well as oral ATTR stabilizers for wild type ATTR cardiac amyloidosis are available [1]. These promising agents can bring transthyretin deposition within the myocardium to a hold. However, a reduction in the left ventricular deposition and function loss was not seen, underlining the importance of an early diagnosis. Already, the addition of global left ventricular strain by the use of speckle tracking offers incremental diagnostic value; in particular, the distinct pattern of ‘apical sparing’ is suggestive of amyloid deposition [37]. Since atrial amyloidosis is characterized by a significant loss of atrial function, atrial reservoir function measurement might further increment the diagnostic value of echocardiography, especially in early stages where ventricular deposition is not yet very advanced.

Reduced left atrial function is associated with cerebrovascular events independent of atrial fibrillation [38]. Therefore, one might hypothesize that severely declined left atrial function might warrant antithrombotic therapy. Recent studies in amyloidosis patients have shown that left atrial dysfunction was associated with all-cause mortality in both ATTR- and AL cardiac amyloidosis [21,28,39]. To date, the incidence of thrombotic events in relation to atrial function in ATTR- and AL cardiac amyloidosis is unknown. A larger, prospective study of atrial function in cardiac amyloidosis to unravel the relation with subsequent thromboembolic events could point out the therapeutic consequences of atrial function measurement.

## 5. Limitations

The current retrospective study can be considered as hypothesis-generating. For the detection of atrial fibrillation, we used ECG, which may be missing atrial fibrillation cases since long-term Holter or implantable loop recordings were not used. While interesting and clinically relevant, the number of (thromboembolic) events during follow-up was too small to study the relationship between the atrial strain at this moment. **We cannot rule out a possible interaction between the given therapy, especially for AL patients, currently receiving active treatment or being in remission. However, once amyloid is deposited, it is unlikely that this will have been cleared in a follow-up time of about 2 years during our study.**

## 6. Conclusions

Compared with AL cardiac amyloidosis, ATTR cardiac amyloidosis patients had lower left atrial reservoir function. This association was independent of the left atrial volume ejection fraction and global longitudinal strain. These findings warrant further investigation into the interplay between restrictive atrial dysfunction and the occurrence of atrial fibrillation. Indeed, it could be a valuable clinical risk assessment tool to predict cardiac amyloidosis patients in sinus rhythm at risk for developing atrial fibrillation, and potentially individuals with an increased risk of thromboembolism while in sinus rhythm.

## Figures and Tables

**Table 1 biomedicines-10-01768-t001:** The baseline characteristics for all immunoglobulin light chain and transthyretin cardiac amyloidosis patients.

	All Patients *N* = 53	AL *N* = 29	ATTR *N* = 24	*p*-Value *
Clinical Variables				
Age (y)	70 ± 8.0	65 ± 7.2	75 ± 5.6	<0.001
Male	44 (83%)	22 (76%)	22 (92%)	0.13
BMI (kg/m^2^)	25 ± 3.3	25 ± 3.7	26 ± 2.5	0.16
DM	6 (11%)	2 (7%)	4 (17%)	0.26
HT	23 (43%)	11 (38%)	12 (50%)	0.38
AF	36 (68%)	16 (55%)	20 (83%)	0.03
NYHA II and III	40 (75%)	22 (76%)	18 (75%)	0.94
MAYO/Gillmore II and III	52 (98%)	28 (97%)	24 (100%)	0.36
NTpro-BNP (µg/mL)	4256 ± 4565	5577 ± 5848	2809 ± 1724	0.04
Troponin I (mg/L)	61 (102)	82 (149)	46 (46)	0.30

* *p*-value between the immunoglobulin (AL)- and transthyretin (ATTR) cardiac amyloidosis (CA) patients. Categorical variables: number (%) and continuous variables with a normal distribution: mean ± SD; with nonparametric distribution: median (IQR). Chi square, independent sample T test and Mann–Whitney U test were used, respectively. BMI: body mass index; DM: diabetes mellitus; HT: hypertension; AF: atrial fibrillation; NYHA: New York Heart Association; BNP: brain natriuretic peptide.

**Table 2 biomedicines-10-01768-t002:** A comparison of the echocardiographic morphological variables between the immunoglobulin light chain and transthyretin cardiac amyloidosis patients.

	All Patients *N* = 53	AL *N* = 29	ATTR *N* = 24	*p*-Value *
Echocardiographic Morphological Variables				
IVSD (mm)	17 (4)	16 (5)	18 (6)	0.006
LV mass index (g/m^2^)	137 ± 40	124 ± 36	156 ± 40	0.006
LA dimension (mm)	44 ± 5.9	43 ± 5.9	45 ± 5.9	0.20
LA maximal volume (mL)	108 ± 39	102 ± 35	114 ± 43	0.27
LA minimal volume (mL)	79 ± 30	76 ± 31	83 ± 28	0.42
LAVI (mL/m^2^)	53 (29)	50 (31)	56 (24)	0.62
RA area (mm^2^)	22 ± 6.4	23 ± 6.9	22 ± 6.0	0.63

* *p*-value between immunoglobulin (AL)- and transthyretin (ATTR) cardiac amyloidosis (CA) patients. Continuous variables with a normal distribution: mean ± SD; with nonparametric distribution: median (IQR). Independent sample T test and Mann–Whitney U test were used, respectively. IVSD: interventricular septum dimension; LV: left ventricle; LA: left atrium; LAVI: left atrial volume index; RA: right atrium.

**Table 3 biomedicines-10-01768-t003:** A comparison of the echocardiographic functional variables between the immunoglobulin light chain and transthyretin cardiac amyloidosis patients.

	All Patients *N* = 53	AL *N* = 29	ATTR *N* = 24	*p*-Value *
Echocardiographic Functional Variables				
LVEF (%)	50.5 (8)	50 (10)	52 (9)	0.87
E/e’	16 (8.6)	16 (6.5)	17 (11)	0.47
TR max (m/s)	2.3 ± 0.6	2.2 ± 0.6	2.4 ± 0.7	0.57
GLS (%)	−11.1 ± 3.5	−11.6 ± 3.9	−10.4 ± 2.7	0.21
LA EF	26 ± 13	26 ± 12	25 ± 14	0.87
LA reservoir strain	9.4 (10)	13.6 (14.7)	7.4 (6.2)	0.017
LA pump strain	5.8 (7.5)	7.5 (7.4)	5.0 (5.6)	0.045
RA reservoir strain	20 (19)	23 (18)	19 (18)	0.33
RA pump strain	13 (13)	13 (14)	10 (9)	0.49

* *p*-value between immunoglobulin (AL)- and transthyretin (ATTR) cardiac amyloidosis (CA) patients. Categorical variables: number (%) and continuous variables: mean ± SD or median (IQR). Chi square, independent sample T test and Mann–Whitney U test were used respectively. LVEF: left ventricle ejection fraction; TR: tricuspid regurgitation; GLS: global left ventricular strain; LA EF: left atrial ejection fraction; RA: right atrium.

**Table 4 biomedicines-10-01768-t004:** The association of variables with the type of cardiac amyloidosis using logistic regression analysis.

Univariable Logistic Regression			
	B-Value	95% Confidence Interval	*p*-Value
LAVI	1.61	0.37–7.00	0.52
LAEF	1.00	0.95–1.04	0.87
LVEF	1.19	0.06–22.7	0.91
GLS	0.90	0.76–1.06	0.21
LA reservoir	0.38	0.15–0.97	0.04
**Multivariate logistic regression**			
LA reservoir	0.16	0.03–0.79	0.02

LAVI: left atrial volume index; LAEF: left atrial ejection fraction; LVEF: left ventricular ejection fraction; GLS: global longitudinal strain. LA reservoir: left atrial reservoir function. LA reservoir, LAVI and LVEF were log transformed because of non-linear distribution.

## Data Availability

The data presented in this study are available on request from the corresponding author.

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
