# Peer review of "Restrictive Atrial Dysfunction in Cardiac Amyloidosis: Differences between Immunoglobulin Light Chain and Transthyretin Cardiac Amyloidosis Patients"

_biomedicines, 2022, doi:10.3390/biomedicines10081768_

Round 1

Reviewer 1 Report

The manuscript topic is actual and important, as it deals with atrial dysfunction in cardiac amyloidosis, seeks for differences in ATTR and AL amyloidosis, hypothesizing that ATTR amyloidosis pts have worse atrial function. Having in mind difficult identification of amyloidosis in early stages, poor prognosis of the pts, high prevalence of atrial fibrillation and a high thromboembolic event rate, the scientific research on atrial function in pts with amyloidosis, which could help to   search for pts at risk for developing atrial fibrillation as well as with an increased risk of thromboembolism while in sinus  rhythm is up to date and of great clinical value.

The Introduction is consecutive, the Materials and Methods as well as Results are well written, with adequate illustrations. The method used- echocardiography with strain imaging is up to date.

 However a few questions appeared:

1).Were the pts in the sinus rhythm during echocardiographic investigation or not? As the pump strain of the atrium was evaluated, it looks obvious that the rhyhtm was sinus, but it should be stated in the Materials and Methods.

2). Can the different age of the pts groups have an impact on the atrial function? It would be preferable to know the authors opinion.

3). What was the treatment for the pts applied?

Do all the pts receive medications during the investigation or none of them were treated? Maybe pts with AL amyloidosis were under the treatment or in the remission phase? The treatment question, which may have influence on the investigation data, should be clearly stated.

The discussion is consistent and thorough.

The English level of the manuscript is high. The manuscript will be interesting to the wide auditorium of medical specialists, especially cardiologists, hematologists, neurologists, etc., dealing with amyloidosis pts.

I highly recommend the manuscript for publication, owing to very clinically relevant topic, presenting new data and promoting new topics for further research.

Reviewer 2 Report

This manuscript investigates the differences in atrial function between individuals diagnosed with either AL or ATTR amyloidosis, as assessed by echocardiography. Consistent with previous studies, but using a new technique in a small cohort, the manuscript shows that patients with ATTR amyloidosis have decreased left atrial function, which the authors propose is the cause of their increased risk of atrial fibrillation, as compared with patients with AL amyloidosis.

The manuscript provides important data describing the differences between the two most common forms of cardiac amyloidosis. These differences are important both for understanding the underlying biology of the diseases and may potentially contribute to differential diagnosis. It is technically sound and suitable for publication with minor corrections.

I have a few suggestions that the authors may consider addressing to strengthen the manuscript.

1.       The manuscript identifies early diagnosis as a way to improve cardiac amyloidosis outcomes. However, the data focuses on the difference between AL and ATTR amyloidosis, rather than identifying either type of amyloidosis at presentation. At a minimum, this should be made more clear. However, the authors could also ask whether their measurements could differentiate between amyloidosis and other forms of heart failure by including literature-derived values for healthy individuals and those with a more common form of cardiomyopathy in tables 1-4. If available, these data would give helpful context.

2.       The differences between AL and ATTR could be due to other factors such as the age of the patients and their degree of heart failure. This is inevitable with small rare-disease cohorts but should be made clear in the text.

3.       Given the small dataset, it would be helpful to plot the individual datapoints for a few key measured variables to provide a better impression for the reader of the signal to noise.

4.       Minor language points:

a.       Line 98: “left atrial fibrosis” – does this refer to non-amyloid fibrosis or amyloid deposition?

b.       Line 99: “substrate” – clarify whether fibrosis is causative, diagnostic or only correlated with fibrillation

c.       Line 198: “excluded because left atrial strain was technically unsuccessful” – measurement of left atrial strain?

d.       Line 276: “Off course” – “of course”?
